# Domain Knowledge Enhanced Vision-Language Pretrained Model for Dynamic Facial Expression Recognition

Liupeng Li
Zhejiang Lab
Hangzhou, China
liliupeng@zhejianglab.com

Yuhua Zheng
Zhejiang Lab
Hangzhou, China
zhengyh@zhejianglab.com

Shupeng Liu
Shanghai University
Shanghai, China
liusp@shu.edu.cn

Xiaoyin Xu
Zhejiang University
Hangzhou, China
xiaoyinxu@zju.edu.cn

Taihao Li*
Hangzhou Institute for Advanced
Study, UCAS
Hangzhou, China
lith@ucas.ac.cn

## Abstract

Dynamic facial expression recognition (DFER) is a rapidly developing field that focuses on recognizing facial expressions in video sequences. However, the complex temporal modeling caused by noisy frames, along with the limited training data significantly hinder the further development of DFER. Previous efforts in this domain have been limited as they tackled these issues separately. Inspired by recent advances of pretrained vision-language models (e.g., CLIP), we propose to leverage it to jointly address the two limitations in DFER. Since the raw CLIP model lacks the ability to model temporal relationships and determine the optimal task-related textual prompts, we utilize DFER-specific domain knowledge, including characteristics of temporal correlations and relationships between facial behavior descriptions at different levels, to guide the adaptation of CLIP to DFER. Specifically, we propose enhancements to CLIP's visual encoder through the design of a hierarchical video encoder that captures both short- and long-term temporal correlations in DFER. Meanwhile, we align facial expressions with action units through prior knowledge to construct semantically rich textual prompts, which are further enhanced with visual contents. Furthermore, we introduce a class-aware consistency regularization mechanism that adaptively filters out noisy frames, bolstering the model's robustness against interference. Extensive experiments on three in-the-wild dynamic facial expression datasets demonstrate that our method outperforms the state-of-the-art DFER approaches. The code is available at https://github.com/liliupeng28/DK-CLIP.

## CCS Concepts

• **Computing methodologies** → **Artificial intelligence**.

*Taihao Li is the corresponding author

## Keywords

Dynamic Facial Expression Recognition; Vision-Language Model

**ACM Reference Format:**
Liupeng Li, Yuhua Zheng, Shupeng Liu, Xiaoyin Xu, and Taihao Li. 2024. Domain Knowledge Enhanced Vision-Language Pretrained Model for Dynamic Facial Expression Recognition. In *Proceedings of the 32nd ACM International Conference on Multimedia (MM '24), October 28-November 1, 2024, Melbourne, VIC, Australia*. ACM, New York, NY, USA, 10 pages. https://doi.org/10.1145/3664647.3681708

## 1 Introduction

Facial expression recognition (FER) has garnered significant attention in the computer vision community due to its crucial role in various applications, such as human-computer interaction (HCI) [5, 7, 26], driver or student status monitoring [14, 22], and medical diagnosis [2, 17]. According to the type of input data, FER can be categorized into static FER (SFER) and dynamic FER (DFER). SFER focuses on static facial images, while DFER deals with dynamic image sequences. The continuity of human emotional expression implies that dynamic image sequences offer richer information regarding the genuine emotional states, thereby facilitating more accurate emotion recognition. Consequently, DFER receives increasing attentions owing to its significance in practical applications.

Previous works in DFER primarily focus on capturing temporal correlations within video frames using approaches based on 3DCNNs [4, 15], RNNs [1, 11, 49] and Transformers [20, 25, 28, 50]. Although these methods have achieved notable performance, the issues of complex temporal correlations and scarce training data still hinder the progress of DFER. Firstly, the existence of non-target frames complicates the modeling of temporal correlations in DFER, distinguishing it from general video understanding tasks. As shown in Fig. 1 (a), dynamic facial sequences often exhibit a transition between target and non-target emotions. Meanwhile, current facial expression datasets typically offer only video-level labels, lacking snippet-level annotations with precise temporal locations. Models trained with this type of supervision are susceptible to producing inaccurate predictions owing to the disruptive influence of noisy frames. Recently, researchers have attempted to address the complex temporal correlations modeling in DFER. NR-DFERNet [21] mitigates the impact of noisy frames by introducing a dynamic class token and a snippet-based filter. M3DFEL [39] proposes mod-

Liupeng Li, Yuhua Zheng, Shupeng Liu, Xiaoyin Xu, and Taihao Li

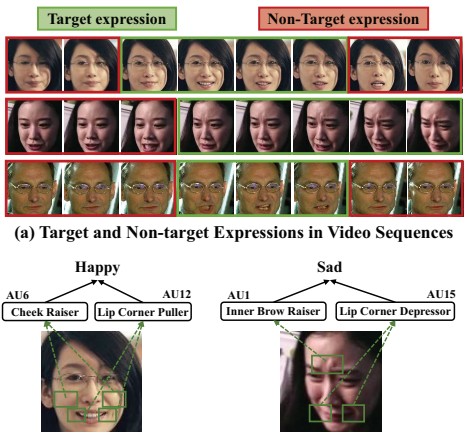

**(a) Target and Non-target Expressions in Video Sequences**

**(b) Relation between Facial Expression and Action Unit (AU)**

**Figure 1: In-the-wild dynamic facial expressions. (a) In the context of video sequences, the non-target expressions constitute noisy frames, which present notable challenges DFER. (b) Facial action units (AUs) serve as descriptors of local facial behaviors associated with particular expressions.**

eling DFER as a multi-instance learning problem, where non-target frames are disregarded while target frames are emphasized. Secondly, the prohibitive cost of annotating facial expression data limits the availability of training data for previous DFER methods, thereby restricting their further advancement. MARLIN[3] and VideoMAE [34] leverage large-scale self-supervised pretraining on unlabeled facial videos to address the constraints posed by the limited training data in existing datasets. Despite their state-of-the-art performance, these methods exhibit notable limitations, including substantial pretraining costs, task-independent representations, and ignoring the impact of noisy frames. Most recently, the advancements in adapting vision-language pretraining models, such as CLIP [31], to various downstream computer vision tasks [29, 52, 53] demonstrate their remarkable transfer and generalization abilities. CLIPER [19] and DFER-CLIP [51] finetune the pretrained CLIP model to promote the development of DFER, without pretraining on unlabeled facial videos. Although these CLIP-based methods have achieved excellent DFER performance, they still employ simple temporal average pooling or single-layer Transformers for feature aggregation across frames, ignoring the detrimental impact of noisy frames on learning robust temporal facial features. In addition, CLIPER necessitates a pretraining phase to learn multiple text descriptors for expressions, whereas DFER-CLIP entails carefully guiding large language models (LLMs) to initially produce descriptions of facial behaviors associated with expressions. The process of training or generating text descriptions is not only time demanding but also fails to ensure the model's generalization across diverse visual samples.

To jointly address the issues of noisy frames and limited training data, we suggest utilizing the domain knowledge in DFER, including characteristics of temporal correlations and relationships between facial behavior descriptions at different levels, to effectively guide CLIP in adapting to DFER. In other words, we incorporate DFER-specific **D**omain **K**nowledge into the **CLIP** model, dubbed **DK-CLIP**, to further promote its performance for the DFER task. To tackle the issue of noisy frames, we present a hierarchical

video encoder equipped with a cross-frame attention module to capture short-term temporal correlations in local snippets and a multi-snippets integration module to model long-term temporal correlations among these snippets. Subsequently, we introduce a class-aware consistency regularization mechanism aimed at mitigating the detrimental effects of noisy frames and bolstering the model's robustness through a noise reducing module and a consistency loss. As for the challenge of limited training data, our method aligns with existing works [19, 51], reducing the need for labeled data by fine-tuning pretraining parameters. Furthermore, since the integration of expression class names with textual descriptions of facial action units (AUs) [9] offers semantic information that is highly pertinent to visual facial content (as illustrated in Fig. 1 (b)), we leverage the descriptions of AUs to enrich the semantic content of text prompt for basic expressions. Additionally, we condition the learning of textual prompts on visual features using a cross-modal attention module, enabling the model to adapt to diverse sample variations. To verify the effectiveness of DK-CLIP, we conduct experiments on three in-the-wild datasets (DFEW [15], FERV39K [41], MAFW [24]). The results demonstrate that the proposed DK-CLIP outperforms the state-of-the-art methods, highlighting its ability to mitigate the effects of limited training data and noisy frames.

In summary, our key contributions are as follows:

- We introduce DK-CLIP, a novel CLIP-based approach that jointly addresses the limitations posed by insufficient training data and noisy frames in DFER.
- DK-CLIP not only offers an effective hierarchical video encoder to modeling complex temporal correlations amidst noise interference, but also harnesses prior knowledge to steer task-related textual prompt generation. With the incorporation of the class-aware consistency regularization mechanism, DK-CLIP is also applicable for fine-grained expression analysis and expression intensity prediction in a weakly supervised setting.
- We conduct extensive experiments on three DFER datasets, and our proposed DK-CLIP achieves state-of-the-art results compared with other methods.

## 2 Related Works

### 2.1 Dynamic Facial Expression Recognition

Deep learning has significantly promoted the advancement of DFER. Early works [11, 15, 18, 49] typically employ RNNs and 3DCNNs to model the temporal relationships among frames. Furthermore, several studies [28, 50] exploit the global dependency modeling ability of Transformer to extract spatial-temporal features from facial video sequences. However, these methods exhibit limited performance since they ignore the varying expression intensities and noisy frames within dynamic facial sequences. Recently, IAL [20] introduces an intensity-aware loss, enabling the network to distinguish frames based on their expression intensities and pay extra attention to low-intensity sequences. NR-DFERNet [21] employs a dynamic class token and a snippet-based filter to mitigate the impact of target irrelevant frames during temporal modeling and the decision-making stage, respectively. Additionally, M3DFEL [39] formulates DFER as a multi-instance learning problem, treating each video as a bag containing target and non-target instances, and

devises a dynamic long-term instance aggregation module to learn the long-term temporal relationships and dynamically aggregate the instances. Although these methods have achieved good performance, they are limited by the insufficient training samples in the existing DFER datasets.

To address the challenge of limited training samples, large-scale self-supervised pretrained methods have been proposed. For instance, MARLIN [3] and MAE-DFER [34] pretrain masked autoencoders on massive non-annotated facial videos, and then finetune them for DFER. The self-supervised pertained models exhibit notable limitations, including substantial pretraining costs and task-independent representations. Furthermore, CLIPER [19] and DFER-CLIP [51] directly adapt the pretrained CLIP to DFER, enabling task-related representation learning and obviating the need for additional pretraining processes. However, these models primarily learn temporal dependencies through video masking autoencoders (VideoMAEs [36]) or temporal average pooling, struggling to model complex temporal dependencies in the presence of noisy frames.

Unlike existing CLIP-based methods for DFER, we propose leveraging unique domain knowledge in DFER to enable CLIP to handle complex temporal dependencies, thereby simultaneously addressing the issues of noise frames and limited training data.

## 2.2 Vision-Language Pretraining

Vision-language pretraining [48] has demonstrated remarkable generalization capabilities to various downstream computer vision tasks. One of the most representative works is CLIP [31], which leverages contrastive learning to jointly learn image-language representations from a large-scale dataset, comprising 400 million image-text pairs sourced from the web. CLIP's remarkable transferability has propelled advancements in various downstream tasks [16, 27, 40, 43]. A critical challenge in adapting the pretrained CLIP for downstream visual tasks is prompt engineering, which is highly time-consuming due to the substantial impact of minor wording variation on performance. CoOp [53] first pioneers the application of context optimization for automatic prompt engineering, introducing learnable prompts for textual inputs, rather than relying on hand-crafted templates. The follow-up CoCoOp [52] utilizes visual feature to generate input-conditional prompts, thereby enhancing generalization abilities. In addition, the unsuitability of the vision encoder devised for images in handling videos potentially restrict CLIP's application in video understanding tasks. X-CLIP [29] introduces a cross-frame attention mechanism to capture long-range frame dependencies along the temporal dimension, enabling CLIP to process videos as input. Moreover, CLIP-VIP [44] facilitates CLIP's ability to handle both images and videos via a proxy-guided video attention mechanism. In addition to the modifications suggested by existing methods, we further leverage domain knowledge to inform the design of video encoders and the learning of textual prompts, enhancing CLIP's adaptability to DFER.

## 2.3 Facial Action Units

Facial action units (AUs) [10, 35] encode subtle facial behaviors commonly observed during the production of facial expressions. For instance, AU4, which describes the facial behavior of lowering eyebrows, is often associated with expressions of *sad*, *fear*,

and *angry*. Actually, the categories of facial expressions provide holistic descriptions of facial behaviors, whereas AUs capture local variations on faces. The pioneering work [9] comprehensively summarizes the relationships between expressions and AUs, offering prototypical AUs for each basic expression. Given that AUs offer more detailed information about different expressions, several studies [6, 23, 30, 45] utilize them to enhance FER performance. Nonetheless, owing to the costly and labor-intensive nature of AU annotation, these studies often rely on prior relationships between AUs and expressions as supplementary supervision signals during model training. In our study, we utilize the textual descriptions of relevant AUs to enrich the semantic content associated with each expression class name.

## 3 Methodology

### 3.1 Revisiting CLIP

CLIP [31] comprises an image encoder and a text encoder to extract image and text features, respectively. After obtaining visual and textual features, CLIP learns a joint encoding space for both modalities using a contrastive loss based on the cosine similarity between the two features. Consequently, the prediction process of the image classification task based on the pretrained CLIP can be formulated as follows:

$$p(y|\boldsymbol{x}) = \frac{\exp(\text{sim}(\boldsymbol{x}, \boldsymbol{c}_y)/\tau)}{\sum_{i=1}^{K} \exp(\text{sim}(\boldsymbol{x}, \boldsymbol{c}_i)/\tau)}, \tag{1}$$

where $\text{sim}(\cdot, \cdot)$ denotes cosine similarity, $K$ represents the number of classes, and $\tau$ is a temperature parameter. The image feature $\boldsymbol{x}$ and the textual descriptions that specify the class $\boldsymbol{c}_y$ are extracted using the image encoder and text encoder of the pretrained CLIP. Our work is focused on effectively adapting CLIP for DFER through the design of an efficient video encoder and learnable textual prompts.

### 3.2 DK-CLIP: Overview

As shown in Fig. 2, our proposed method comprises visual feature extraction, textual feature extraction, and prediction. In the visual feature extraction part, facial expression sequences, consisting of $T$ frames, are first embedded into patch embeddings, which are subsequently divided into $N$ snippets. Following this, the cross-frame attention module extracts local visual features, while the multi-snippet integration module extracts global visual features. In the textual feature extraction part, the class names of expressions are aligned with AUs based on prior knowledge summarized from previous work [9]. Subsequently, the cross-modal attention module takes textual features and noise-free visual features as inputs to learn textual prompts tailored for DFER. In the prediction part, the noise reducing module is employed to erase noisy local snippets. Then, all extracted visual features, excluding noisy ones, are utilized to calculate similarity with textual prompts for classification and engage in consistency regularization. We will elaborate on these components in the subsequent sections.

### 3.3 Visual Feature Extraction

The primary challenge in adapting CLIP to DFER lies in how to learn robust and discriminative visual features through effectively

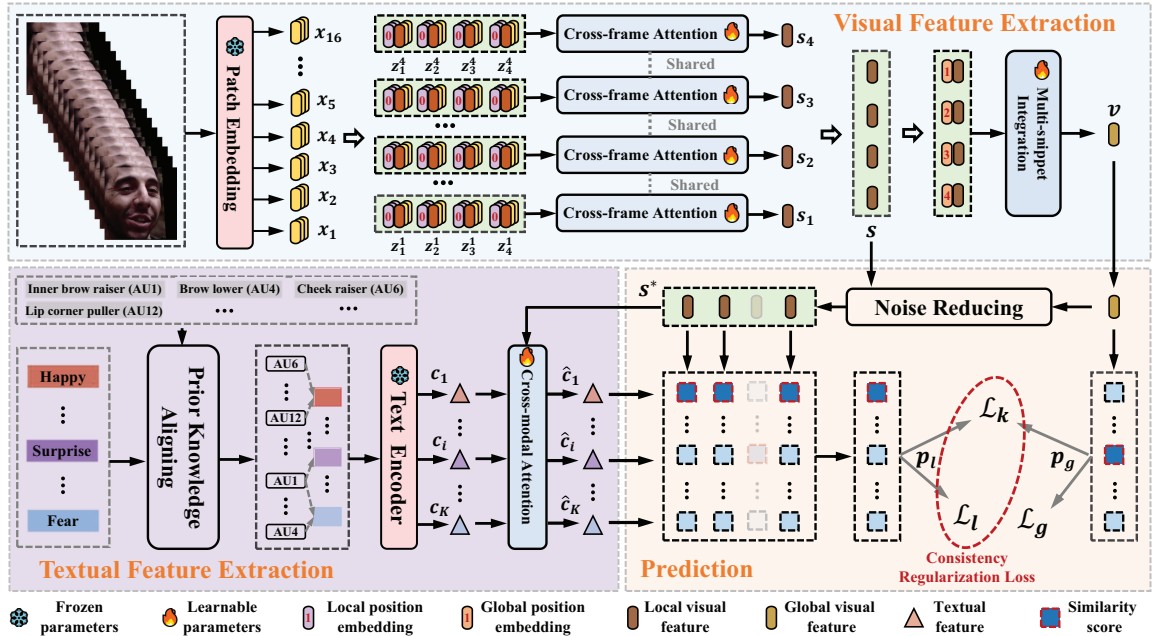

**Figure 2: An overview of our proposed DK-CLIP. In the visual feature extraction part, the cross-frame attention module and multi-snippet integration module are responsible for extracting local and global visual features, respectively. In the textual feature extraction part, class names of expressions are initially aligned with corresponding AUs based on prior knowledge. Subsequently, the textual features undergo enhancement through the integration of noise-free visual features. During the prediction stage, the noise reducing module and consistency regularization mechanism work together to adaptively mitigate the influence of noisy visual features. Notably, the consistency regularization is only employed during the training process.**

modeling the temporal correlations within video frames. Nonetheless, the presence of noisy frames requires consideration of both short- and long-term temporal dependencies when model temporal correlations in DFER, which differentiates from general video recognition task. Inspired by the work [29] that employs a cross-frame attention mechanism to capture temporal correlations in videos, we develop a hierarchical video encoder that first extracts local features from local sequences, and then extracts global features.

Specifically, given a video clip $V \in \mathbb{R}^{T \times H \times W \times 3}$, consisting of $T$ sampled frames with $H$ and $W$ denoting the spatial resolution, we follow the Vision Transformer (ViT) [8] to divide the $t$-th frame into $M$ non-overlapping patches and subsequently embed them into patch embeddings, denoted as $\{x_{t,i}\}_{i=1}^{M}$, where $t \in \{1, \cdots, T\}$ denotes the absolute temporal index. Then, the patch embeddings of $T$ frames are evenly divided into $N$ snippets. To capture temporal correlations within each local snippet, we incorporate a cross-frame attention module (CFAM) composed of an $L$-layer Transformer. The input to the CFAM at frame $\hat{t}$ in the $n$-th snippet is denoted as:

$$z_{\hat{t}}^{n} = [x_{\hat{t},c}^{n}, x_{\hat{t},1}^{n}, x_{\hat{t},2}^{n}, \cdots, x_{\hat{t},M}^{n}] + e^{spa}, \quad (2)$$

where $x_{\hat{t},c}^{n}$ is the learnable *class* token, $\hat{t}$ is the relative temporal index in local snippets, and $e^{spa}$ represents the spatial position encoding. Subsequently, we utilize the *message* token $m_{\hat{t}}^{n}$, linearly transformed from $x_{\hat{t},c}^{n}$, to abstract and exchange visual information across frames in each local snippet. Specifically, in the $l$-th layer of the CFAM, the *message* tokens of $n$-th snippet, denoted as

$m^{n} = [m_1^{n}, m_2^{n}, \cdots, m_{\hat{T}}^{n}]$, are utilized to capture spatio-temporal relationships among local frames in the local snippet, which is formulated as:

$$\tilde{m}^{n} = m^{n} + \text{MHSA}(\text{LN}(m^{n})), \quad (3)$$

where LN indicates layer normalization and MHSA represents multi-head self-attention [38]. Subsequently, $z_{\hat{t}}^{n}$ and $\tilde{m}_{\hat{t}}^{n}$ are concatenated to diffuse spatio-temporal dependencies across each frame, which defined as following:

$$[\hat{z}_{\hat{t}}^{n}, \hat{m}_{\hat{t}}^{n}] = [z_{\hat{t}}^{n}, \tilde{m}_{\hat{t}}^{n}] + \text{MHSA}(\text{LN}([z_{\hat{t}}^{n}, \tilde{m}_{\hat{t}}^{n}])), \quad (4)$$

where $[\cdot, \cdot]$ concatenates the frame tokens and message tokens. Ultimately, a snippet-level representation $s_n$ is derived by:

$$s_n = \frac{1}{\hat{T}} \sum_{\hat{t}=1}^{\hat{T}} \hat{x}_{\hat{t},c}^{n}. \quad (5)$$

As a result, the CFAM outputs a collection of snippet features, denoted as $s = [s_1, s_2, \cdots, s_N]$.

To derive a comprehensive representation for the entire video clip $V$, we introduce a multi-snippet integration module (MSIM) to synthesize local snippet features into a single video-level feature $v$, which is expressed as:

$$v = \text{AvgPool}(\text{FFN}(\text{MHSA}(s + e^{tem}))), \quad (6)$$

where $e^{tem}$ is the relative temporal position embedding, AvgPool denotes the average pooling, and FFN indicates the feed-forward network.

## 3.4 Textual Feature Extraction

Existing works [19, 51] have demonstrated that constructing suitable textual prompts plays a key role in adapting pretrained vision-language models to DFER task. However, the ways they generate textual prompts involve complicated preprocessing steps that typically require a significant amount of time. Therefore, we propose a straightforward approach to enrich textual prompts by combining the class names of expressions and the textual descriptions of relevant AUs, guiding the model to focus on facial regions related to expressions. Furthermore, we leverage visual contents to enhance the constructed textual prompts, enabling the model to learn relevant information for each expression.

**Enriching Textual Prompts with AUs.** Traditional ways of adapting CLIP to recognition tasks often rely on manually designed textual prompts, such as "*A photo of a {class}*" or simply "*{class}*", where "*class*" primarily refers to objects or scenes. However, due to the semantic gap between class names and visual contents, these traditional ways are not suitable for adapting CLIP to DFER. According to existing works [10, 35], it is known that AUs describe facial behaviors closely related to basic expressions (excluding *Neutral*). As suggested in previous literature [9] that each expression is associated with several AUs, we consider aligning basic expressions with the primary relevant AUs to generate more semantically rich textual prompts, as shown in Tab. 1. Specifically, we concatenate the class name with the descriptions of related AUs to construct textual prompts. Meanwhile, we use "or" to indicate the semantics of AUs that are not always activated simultaneously. For instance, the textual prompt for "*Happiness*" is reformulated as "*Happiness with cheek raiser, lip corner puller or lip part.*", where "*cheek raiser*", "*lip corner puller*", and "*lip part*" are the textual descriptions of AU6, AU12 and AU25, respectively. Notably, the textual prompts for expressions without relevant AUs consist solely of the class name.

**Enhancing Textual Prompts with Visual Contents.** Although textual prompts for basic expressions are enriched through AUs, the manually designed prompts remain inadequate for generalizing across varied visual situations. We follow the conditional prompt learning mechanism [29, 52], leveraging visual content information to enhance textual prompts. Specifically, given a set of text embeddings $c = [c_1, c_2, \cdots, c_K]$ derived from enriched textual prompts, we propose a cross-modal attention module (CMAM) to facilitate conditional prompt learning, which is formalized as following:

$$\bar{z} = \frac{1}{T} \sum_{t=1}^{T} z_t, \tag{7}$$

$$\hat{c} = c + \gamma \cdot \text{FFN}(\text{MHSA}(c, \bar{z})), \tag{8}$$

where $\hat{c}$ represents the enhanced prompts, $\gamma$ denotes a learnable parameter initialized at 0.1, and $z_t$ is the ultimate output of the $t$-th frame obtained from the CFAM, excluding the *class* tokens. In MHSA, we utilize the text representation $c$ as the query and the visual content representation $\bar{z}$ as both the key and value.

## 3.5 Class-aware Consistency Regularization

The previous work [21] has verified that noisy frames significantly impact the performance of DFER methods, while the current datasets lack supervised information for their identification. To mitigate the influence of noisy frames on the visual and textual features,

**Table 1: The prior relationships between basic expressions and the primary relevant AUs. The detailed textual prompts of each expression can be found in the appendix.**

| Basic Expression | Primary Relevant AUs |
|---|---|
| Happiness | AU6, AU12, AU25 |
| Sadness | AU1, AU4, AU15, AU17 |
| Anger | AU4, AU7, AU17, AU24 |
| Surprise | AU1, AU2, AU5, AU25, AU26 |
| Disgust | AU9, AU10, AU17 |
| Fear | AU1, AU2, AU4, AU5, AU20, AU25 |

we introduce a class-aware consistency regularization mechanism, consisting of a noise reducing module and a consistency regularization loss, to enable the model to adaptively perceive and erase noise features.

**Noise Reducing Module.** Due to lacking supervision signals to identify noisy frames, we introduce a simple yet efficient approach to adaptively perceive noisy frames and mitigate their adverse effects on DFER, i.e., Noise Reducing Module (NRM). The NRM perceives and erases the noisy frames by leveraging visual similarities between local snippet features and the holistic feature as indicators. Specifically, we compute the similarity between each snippet feature $s_n$ and the holistic feature $v$, resulting in a set of similarity scores denoted as $a = [a_1, a_2, \cdots, a_N]$. Then, we employ the normalized similarity scores $a^*$ to identify potential noisy snippet features and erase them, which can be defined as follows:

$$s_n^* = \begin{cases} \varnothing, & a_n^* < \theta, \\ s_n, & \text{otherwise,} \end{cases} \tag{9}$$

where $\theta$ represents a pre-determined hyperparameter and $\varnothing$ denotes empty vector. The remained snippet features then constitute a new set $s^*$, containing local features that are less impacted by noisy frames. Additionally, only $T^*$ frames, excluding those identified as noisy, are utilized in computing $\bar{z}^*$. Meanwhile, $\bar{z}$ in Eq. 8 is replaced by $\bar{z}^*$ to mitigate the impact of noisy frames on textual prompts.

**Consistency Regularization Loss.** Due to noisy frames are identified based on local snippet features and holistic feature in an unsupervised manner, the precision of these identified noisy frames cannot be guaranteed in the early training phases. The inconsistency between the two features may result in erasing keyframes incorrectly. Meanwhile, the influence of noisy frames on the overall holistic visual features remain unchanged. Actually, the presence of noisy frames or erasing keyframes is equivalent to introduce perturbations to the sample, potentially resulting in inconsistency between local snippet features and holistic feature. Consequently, we introduce the consistency regularization employed in semi-supervised learning [33, 46] to enforce similar predictions for perturbed versions of the same sample. Specifically, by utilizing the holistic feature $v$ and the text features $\hat{c}$, we can compute the global prediction result $p_g$ using Eq. 1. Additionally, we can derive a set of local prediction results based on the refined snippet features $s^*$. Subsequently, we generate a unified local prediction $p_l$ by fusing local predictions, which can be formulated as:

$$p_l = \sum_{i=1}^{N^*} w_i \cdot p_i, \tag{10}$$

**Table 2: Performance comparison of our DK-CLIP with the state-of-the-art methods on DFEW.**

| Methods | Accuracy of Each Emotion(%) | | | | | | | Metrics(%) | |
|---|---|---|---|---|---|---|---|---|---|
| | Hap. | Sad. | Neu. | Ang. | Sur. | Dis. | Fea. | UAR | WAR |
| FormerDFER [50] | 84.05 | 62.57 | 67.52 | 70.03 | 56.43 | 3.45 | 31.78 | 53.69 | 65.70 |
| STT [28] | 87.36 | 67.90 | 64.97 | 71.24 | 53.10 | 3.49 | 34.04 | 54.58 | 66.45 |
| DPCNet [42] | - | - | - | - | - | - | - | 55.02 | 66.32 |
| NR-DFERNet [21] | 88.47 | 64.84 | 70.03 | 75.09 | 61.60 | 0.00 | 19.43 | 54.21 | 68.19 |
| IAL [20] | 87.95 | 67.21 | 70.10 | 76.06 | 62.22 | 0.00 | 26.44 | 55.71 | 69.24 |
| M3DFEL [39] | 89.59 | 68.38 | 67.88 | 74.24 | 59.69 | 0.00 | 31.63 | 56.10 | 69.25 |
| MARLIN (ViT-B/16) [3] | 85.77 | 66.64 | 67.22 | 69.54 | 60.72 | 0.00 | 27.72 | 53.94 | 66.74 |
| MAE-DFER (ViT-B/16) [34] | 92.92 | **77.46** | 74.56 | 76.94 | 60.99 | 18.62 | 42.35 | 63.41 | 74.43 |
| CLIPER (ViT-B/32) [19] | - | - | - | - | - | - | - | 57.56 | 70.84 |
| DFER-CLIP (ViT-B/32) [51] | 91.12 | 75.34 | 71.15 | 74.09 | 56.30 | 11.72 | 37.81 | 59.61 | 71.25 |
| DK-CLIP (ViT-B/32) | 93.03 | 74.96 | 70.45 | 77.28 | 60.45 | 7.72 | 39.04 | 60.42 | 72.98 |
| DK-CLIP (ViT-B/16) | **94.61** | 76.19 | **75.06** | **78.65** | **63.61** | **23.60** | **42.93** | **64.95** | **75.41** |

where $\boldsymbol{p}_i$ represents the local prediction of $i$-th snippet feature, $N^*$ denotes the number of snippet features in $\boldsymbol{s}^*$, and $w_i$ is fusion weight determined from the visual similarity scores $\boldsymbol{a}^*$. Then, the class-aware consistency regularization between $\boldsymbol{p}_g$ and $\boldsymbol{p}_l$ is achieved via minimizing the following loss function:

$$\mathcal{L} = \mathcal{L}_g + \lambda \cdot (\mathcal{L}_l + \mathcal{L}_k), \qquad (11)$$

where $\mathcal{L}_g$ and $\mathcal{L}_l$ represent the cross-entropy loss of $\boldsymbol{p}_g$ and $\boldsymbol{p}_l$, respectively. $\mathcal{L}_k$ denotes the KL-loss [13], which measures the KL-Divergence between $\boldsymbol{p}_g$ and $\boldsymbol{p}_l$. $\lambda$ is a scalar hyperparameter that controls the weight of the consistency regularization loss, which comprises $\mathcal{L}_l$ and $\mathcal{L}_k$. We enforce consistent prediction outputs for the same sample across varying perturbations by aligning predicted categories and probability distributions. This not only enhances the NRM's capability to perceive noisy frames but also mitigate the influence of noisy features on global feature.

## 4 Experiments

### 4.1 Experimental Setup

**Databases.** We conduct experiments on three in-the-wild datasets, including DFEW [15], FERV39K [41], and MAFW [24]. DFEW consists of 16,372 video clips collected from more than 1,500 movies, and each video clip is annotated with seven basic expressions, including *Happiness*, *Sadness*, *Neutral*, *Anger*, *Surprise*, *Disgust*, and *Fear*. FERV39K comprises 38,935 video clips collected from different scenarios, and each video clip is assigned to one of the seven primary expressions as in DFEW. MAFW is a multi-modal compound affective database with 10,045 video clips. Each video clip is annotated with 11 compound expressions (containing *Contempt, Anxiety, Helplessness, Disappointment*, and seven basic expressions) and a textual description of the subject's affective behavior. We only use the video modality to evaluate our proposed method. DFEW and MAFW provide data partitioning settings for 5-fold cross-validation, while FERV39K splits all data into training and test sets without overlapping. To ensure a fair comparison with other methods, we conduct experiments on the default setting provided by each dataset.

**Implementation Details.** Our proposed method is based on the pretrained CLIP that uses ViT [8] as the image encoder and a transformer model [32] as the text encoder. Specifically, the patch

**Table 3: Performance comparison of our DK-CLIP with the state-of-the-art methods on FERV39K and MAFW.**

| Methods | FERV39K | | MAFW | |
|---|---|---|---|---|
| | UAR | WAR | UAR | WAR |
| FormerDFER [50] | 37.20 | 46.85 | 31.16 | 43.27 |
| NR-DFERNet [21] | 33.99 | 45.97 | - | - |
| T-ESFL [24] | - | - | 33.28 | 48.18 |
| IAL [20] | 35.82 | 48.54 | - | |
| M3DFEL [39] | 35.94 | 47.67 | - | - |
| MARLIN (ViT-B/16) [3] | 35.13 | 46.64 | 34.83 | 48.05 |
| MAE-DFER (ViT-B/16) [34] | 43.12 | 52.07 | 41.62 | 54.31 |
| CLIPER (ViT-B/32) [19] | 41.23 | 51.34 | - | - |
| DFER-CLIP (ViT-B/32) [51] | 41.27 | 51.65 | 39.89 | 52.55 |
| DK-CLIP (ViT-B/32) | 40.76 | 51.58 | 41.17 | 54.93 |
| DK-CLIP (ViT-B/16) | **43.71** | **52.14** | **43.01** | **56.56** |

embedding module and text encoder of our proposed method inherit the weights directly from the pretrained CLIP, while the CFAM partially inherits the weights from the pretrained ViT. The MSIM and CMAM are randomly initialized. All parameters of CFAM, MSIM, and CMAM are adjusted during training, while the parameters of the patch embedding and text encoder remain frozen. Our proposed model is trained for 50 epochs with five warm-up epochs using the AdamW optimizer. The learning rate, the weight decay, and the batch size are set to 2e-6, 0.001, and 8, respectively. We use data augmentation techniques consisting of Fmix [12] and Mixup [47] to increase the diversity of training data. We sample 16 frames for each video ($T = 16$), and divide them into 4 parts as inputs ($N = 4$). Meanwhile, we align only six basic expressions with AUs for all datasets. We implement the proposed method with the Pytorch toolbox and train it on 2 Tesla V100 GPUs. In all experiments, we take unweighted average recall (UAR, i.e., the average accuracy of each class) and weighted average recall (WAR, i.e., accuracy) as metrics.

### 4.2 Experimental Results

To demonstrate the superiority of our proposed DK-CLIP, we compare it with the state-of-the-art methods on three in-the-wild datasets. Since different methods employ distinct scales of ViT (B/16 and

B/32) for feature extraction, we conduct experiments on both model scales.

The average performances of 5-fold cross-validation results on DFEW are presented in Tab. 2. As can be seen, our proposed DK-CLIP achieves superior performance in both UAR and WAR. Specifically, in comparison to the previously best-performing MAE-DFER [34], DK-CLIP achieves an improvement of 1.54% and 0.98% in terms of UAR and WAR, respectively. Additionally, it is noteworthy that DK-CLIP exhibits significant improvement across most expressions. The performance of DK-CLIP surpasses that of CLIPER [19] and DFER-CLIP [51], thus highlighting the effectiveness of our proposed method in adapting CLIP for DFER. In addition, DK-CLIP significantly outperforms M3DFEL[39], which is trained from scratch, by 8.85%/6.16% of UAR/WAR, demonstrating the effectiveness of leveraging pretrained knowledge for DFER. Tab. 3 presents the experimental results obtained on FERV39K and MAFW. Our proposed DK-CLIP achieves the best performance on both datasets. Specifically, DK-CLIP surpasses MAE-DFER [34] by margins of 1.39% UAR and 2.25% WAR on MAFW. Meanwhile, it outperforms DFER-CLIP with advantages of 1.28% UAR and 2.38% WAR. Owing to the diverse scenarios present in FERV39K compared to DFEW and MAFW, the dataset poses a greater challenge, leading to lower recognition accuracy. Concurrently, the improvement in performance resulting from the increased ViT scale is relatively minor compared to other datasets. In summary, the promising results on three in-the-wild datasets demonstrate the strong generalization ability of DK-CLIP in practical scenarios.

## 4.3 Ablation Study

In this section, we conduct ablation studies to investigate the impact of key factors in DK-CLIP. For simplicity, we only report the results of the ViT-B/16 based model on DFEW fold 3.

**Influence of Key Components.** To evaluate the effectiveness of key components in DK-CLIP, we perform experiments with varying model architectures. The baseline model employs average pooling for temporal modeling and incorporates class names as textual prompts. As shown in Tab. 4, the results demonstrate the effectiveness of modeling temporal correlations via the hierarchical video encoder, along with the enhancement of textual prompts through including AUs and visual contents. Our proposed DK-CLIP outperforms the baseline model by 4.09%/3.17% of UAR/WAR. Furthermore, the integration of class-aware consistency regularization enhances performance by 0.74%/0.77% of UAR/WAR, highlighting its effectiveness in reducing the influence of noisy frames.

**Influence of Different Temporal Modeling.** To evaluate the effectiveness of our proposed hierarchical video encoder, we conduct experiments utilizing diverse temporal modeling approaches. As shown in Tab. 5, our proposed hierarchical video encoder consistently outperforms other approaches, showcasing its prowess in modeling intricate temporal correlations within facial expression sequences.

**Influence of Different Textual Prompts.** We conduct experiments to analyze the influence of different textual prompts on the performance of DK-CLIP. As shown in Tab. 6, we observe that the utilization of learnable prompts results in superior performance due to their capacity to learn task-relevant information   throughout the

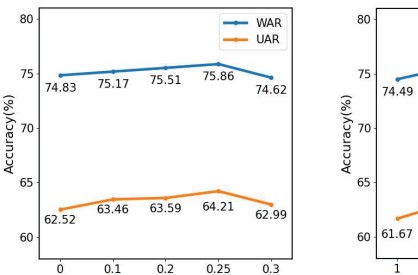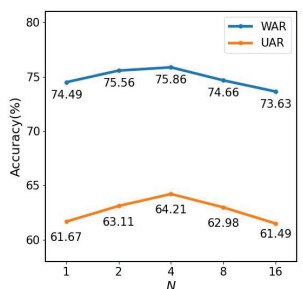

**Figure 3: Ablation study on the noisy frame threshold $\theta$, and the number of local snippets $N$.**

**Table 4: Ablation study on the model architecture. HVE: Hierarchical Video Encoder. ETP: Enhanced learnable Text Prompts. CCR: Class-aware Consistency Regularization.**

| Modules | | | Metrics(%) | |
|---|---|---|---|---|
| HVE | ETP | CCR | UAR | WAR |
| ✗ | ✗ | ✗ | 59.38 | 71.92 |
| ✓ | ✗ | ✗ | 61.43 | 74.83 |
| ✗ | ✓ | ✗ | 61.49 | 73.63 |
| ✓ | ✓ | ✗ | 63.47 | 75.09 |
| ✓ | ✓ | ✓ | **64.21** | **75.86** |

**Table 5: Ablation study on different temporal modeling. (a) Average pooling [19]. (b) Single-layer Transformer [51]. (c) Cross-frame attention [29]. (d) The hierarchical structure.**

| Metrics | Setting(%) | | | |
|---|---|---|---|---|
| | a | b | c | d |
| UAR | 61.49 | 61.67 | 63.47 | **64.21** |
| WAR | 73.63 | 74.49 | 75.09 | **75.86** |

**Table 6: Ablation study on textual prompts. (a) Class names. (b) Learnable prompts [53]. (c) Learnable prompts generated by LLMs [51]. (d) Learnable prompts enriched by all relevant AUs. (e) Learnable prompts enriched by activated AUs.**

| Metrics | Setting(%) | | | | |
|---|---|---|---|---|---|
| | a | b | c | d | e |
| UAR | 62.46 | **64.50** | 62.67 | 63.98 | 64.21 |
| WAR | 74.49 | 75.17 | 75.34 | 75.56 | **75.86** |

training process. Moreover, compared with the learnable prompts generated by LLMs in DFER-CLIP [51], our proposed learnable prompts enhanced by visual content and activated AUs exhibits an improvement of 1.54% and 0.52% in UAR and WAR, respectively. This improvement stems from the fact that the generated textual prompts exhibit significant semantic similarity, whereas our textual prompts, incorporating class names and AUs, exhibit greater discernibility. Furthermore, compared to using "and" in textual prompts to combine AUs that are not always activated simultaneously (Tab. 6 (d)), employing "or" to distinguish their semantics (Tab. 6 (e)) shows better performance.

**Influence of Hyper-parameters.** We investigate the influence of the threshold $\theta$ and the number of local snippets $N$. As shown in

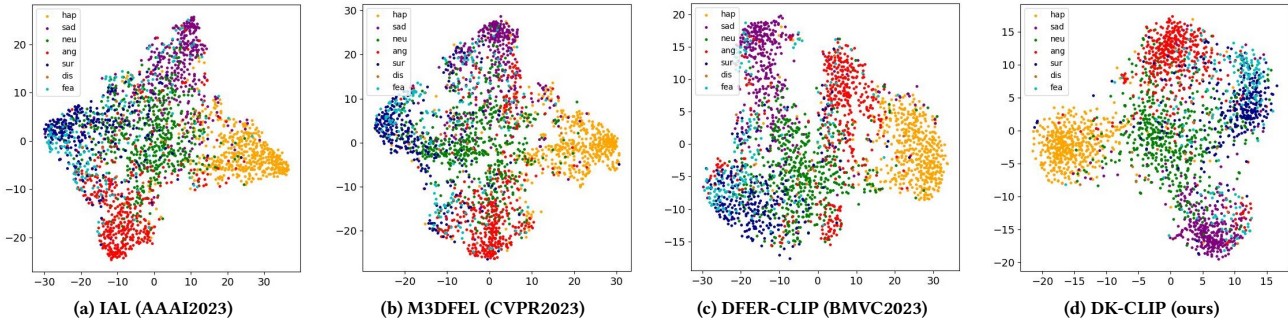

**Figure 4: High-level visual feature visualization on DFEW fold 3 using t-SNE.**

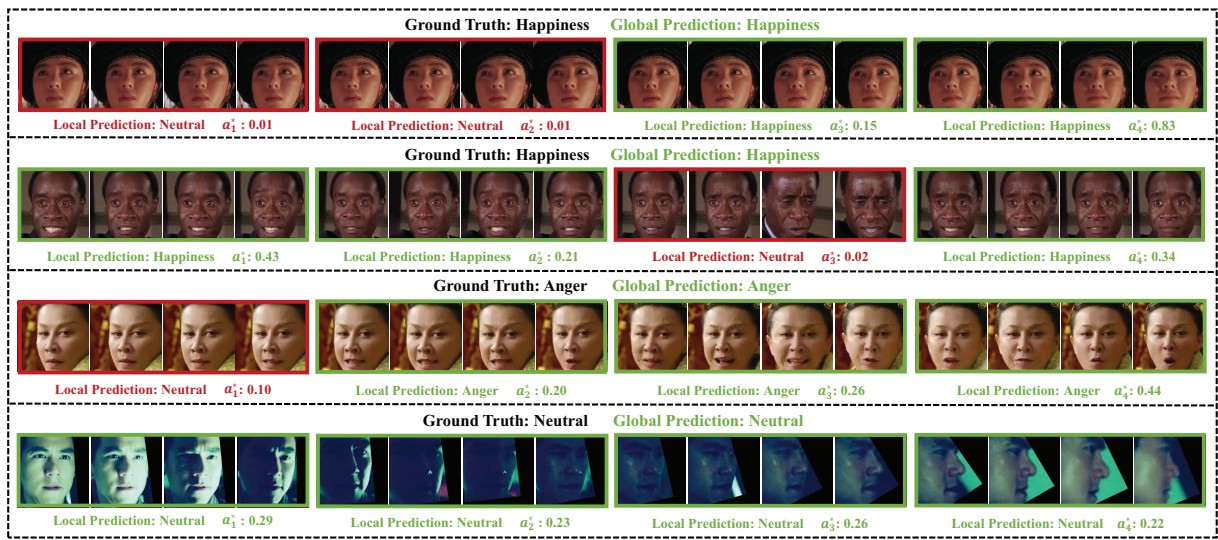

**Figure 5: The recognition results of four video sequences at $\theta = 0.15$. The global and local predictions are obtained by video-level and snippet-level representations, respectively. The local snippets with similarity scores $a^*$ less than $\theta$ are erased.**

Fig. 3, improper setting of noise frame threshold $\theta$ can significantly impact model performance. This is due to the fact that a threshold $\theta$ that is too low fails to effectively erase noise, whereas an excessively high threshold may lead to the erroneous removal of keyframes. Meanwhile, it is evident that setting the number of local snippets to 4 represents an optimal selection for a sampling rate of 16.

## 4.4 Visualization

**Feature Visualization.** To demonstrate the superiority of DK-CLIP compared to state-of-the-art methods, we visualize the learned visual embedding using t-SNE [37] on DFEW fold 3. As can be seen in Fig. 4, the embeddings learned by our method exhibit greater compactness and separability compared to those of other methods, which demonstrates the ability of DK-CLIP to learn highly discriminative visual representations for diverse dynamic facial expressions.

**Recognition Results Visualization.** To further evaluate the effectiveness of the proposed method, we provide detailed local and global prediction results for four samples from DFEW fold 3, as shown in Fig. 5. As we can see, our proposed method accurately

identifies noisy frames and produces reliable predictions. Meanwhile, we observe that the similarity scores also serve as indicators for estimating the expression intensity, as exemplified in the third row of Fig. 5. Furthermore, our proposed method demonstrates robustness against interference such as occlusion and low light conditions.

## 5 Conclusion

In this paper, we propose a novel method to promote the development of DFER through effectively adapting CLIP to DFER. We present a hierarchical video encoder designed to jointly model short- and long-term temporal correlations in DFER. Meanwhile, we align facial expressions and AUs using prior relationships to construct semantically rich textual prompts, which are further enhanced with visual contents. Furthermore, we introduce a class-aware consistency regularization mechanism that comprises a noise reducing module and a consistency regularization loss, aiming to reduce the impact of noisy frames and enhance the robustness of the model to interference. The extensive experimental results demonstrate the superiority of our proposed method.

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
