# OpenReview forum: "Domain Knowledge Enhanced Vision-Language Pretrained Model for Dynamic Facial Expression Recognition"
_acmmm.org/ACMMM/2024/Conference — MM2024 Poster_

### Official Review · Reviewer_NxJ2 · 2024-05-11

**Rating:** 3
**Confidence:** 3

**Summary:**

The paper proposes to utilize domain knowledge to optimize the application of the CLIP model by the proposed DK-CLIP method. DK-CLIP addresses two limitations in DFER: lack of temporal modeling ability and lack of optimizing the task-related textual prompts. Specifically, the authors design a hierarchical video encoder to capture the short- and long-term relationship, introduce the prior knowledge of facial action units and categorical emotion class into the feature generation of text, and devise a class-aware consistency regularization mechanism to aptly filter out noisy frames. Extensive experiments on datasets show superior performance over existing methods.

**Strengths:**

The writing is clear and easy to read.

**Limitations:**

The topic of upgrading existing CLIP-based methods for DFER is hot and crucial but the innovation has only some incremental.

1. In the introduction section, no paragraph is written to clarify that how the proposed work or previous CLIP-based works jointly alleviate the issues of limited training data and noisy frames.
2. Line 117: Does the sentence "restricts the availability of training data for DFER models" means the issue of a limited number of training samples? If it is, why are the following sentences  (L121~L124) mentioned and just only mention the methods for tackling noisy frames?
3. Line238: Why do the self-supervised pertained models lack the ability to handle complete temporal dependencies in DFER? There is no mention in related work 2.1. In addition, Transformer-based methods for DFER can also be divided into supervised and unsupervised techniques. So it is unsuitable to write "In addition to supervised learning methods, large-scale self-supervised pre-trained models..." in the first sentence of the second paragraph of related work 2.1.
4. Line 240: Line 240: "to enhance the pre-trained CLIP model" -> enable CLIP the ability of handling complex temporal dependencies?
5. Line 241: It mentions "both issues simultaneously" but only the issue of lacking the ability to handle complex temporal dependencies in DFER involved.

**Suitability:**

2

---

### Official Review · Reviewer_uoHN · 2024-05-24

**Rating:** 4
**Confidence:** 4

**Summary:**

This paper utilizes domain-specific knowledge from DFER, including temporal features and the relationship between facial behavioral descriptions at different levels, to guide CLIP in adapting to DFER. This resolves the complex temporal modeling issue caused by noisy frames and addresses the limitation of training data.

**Strengths:**

The paper introduces the DK-CLIP method based on the CLIP model, addressing two limitations encountered by DFER. By capturing key facial region features and integrating textual cues, it achieves emotion recognition and classification tasks. The paper demonstrates clear logic, detailed content, complete structure, enriched experimental validation. It significantly enhances the accuracy of emotion recognition.

**Limitations:**

1.If the number of non-target frames significantly exceeds the number of target frames in a video sequence, what strategies can be employed?
2.The visual cases chosen in the method all feature frontal or slightly off-center views of the face. If more than half of the face is occluded, it's uncertain whether the recognition accuracy of these video sequences will decrease. However, there is a lack of analysis on failure cases in such scenarios.

**Suitability:**

3

---

### Official Review · Reviewer_cnt4 · 2024-05-25

**Rating:** 4
**Confidence:** 3

**Summary:**

This research introduces a method to deal with data scarcity and noise issues in the DFER field by adapting CLIP to DFER. Including proposing a hierarchical video encoder to jointly model short-and long-term temporal correlations in DFER in the visual part, and aligning facial expressions and AUs using prior relationships to construct semantically rich textual prompts in the textual part. As well as proposing a class-aware consistency regularization mechanism to reduce the impact of noisy frames.

**Strengths:**

The paper is easy to understand and follow. The authors use clear language and provide sufficient background information for readers who may not be familiar with DFER, Vision-Language Pretraining or AUs. The experimental comparison is relatively sufficient.

**Limitations:**

1. The article regarding the input of CMAM contains contradictory statements in several sections; please check for clarity or errors. Line 328 indicates that CMAM accepts global features as visual input, which seems inconsistent with the descriptions in Figure 2 and line 505.
2. The text overly relies on verbal descriptions, and it is suggested to include more formulas to aid reader comprehension. For example, the descriptions from line 396 to 404 could be entirely replaced with formulas, as well as further descriptions of MSIM in Equation 3 and CMAM in Equation 4.
3. The author has not made the models and algorithms discussed in this paper open source. It is hoped that the author will contribute their work to the community.

**Suitability:**

3

---

### Official Review · Reviewer_2E9b · 2024-05-26

**Rating:** 3
**Confidence:** 4

**Summary:**

This paper proposes a dynamic facial expression recognition framework that leverages DFER-specific domain knowledge, including temporal correlation characteristics and the relationships between different levels of facial behavior descriptions, to guide the adaptation of CLIP, which is pre-trained on general images, to DFER.

**Strengths:**

1. The proposed hierarchical video encoder effectively captures both short- and long-term temporal correlations in videos.
2. The introduced prior knowledge of the relationships between expressions and AUs enriches the textual prompts.
3. The proposed noise reducing module adaptively removes non-target noise frames, as visualized convincingly in Figure 5.
4. Extensive experimental comparisons and ablation analyses demonstrate the excellent performance of the proposed DK-CLIP on the DFER task.

**Limitations:**

1. When enriching the textual prompts with AUs, the proposed method takes a relatively straightforward approach, such as "Happiness with cheek raiser, lip corner puller, or lip part." Despite using "or," a concern is that this sentence still includes textual descriptions of all related AUs. In reality, these AUs do not always activate simultaneously, leading to inaccuracies. An effective way to validate this is to replace "or" with "and" and check for changes in the model's performance.
2. The meanings of some elements in Figure 2 are unclear, such as the set of tokens after patch embedding and the corresponding indices. It is recommended to refine these elements to enhance readability.
3. The cross-frame attention module adopts the structure of the ViT from CLIP. Are the parameters of each cross-frame attention module shared? If not, wouldn't this result in an excessive number of parameters?
4. The paper mentions that "All parameters are adjusted during training." Are the parameters of the patch embedding and text encoder also trainable? This seems counterintuitive.
5. The paper mentions that 𝜆 is a scalar hyperparameter that controls the weight of the consistency regularization, but it also seems to control L_𝑙. Could this be a typo?
1. The conclusive analysis in lines 748 to 751 should be in the "Influence of Different Textual Prompts" section.

**Suitability:**

3

---

### Meta-Review · Area_Chair_WHaf · 2024-07-05

**Recommendation:** Accept (Poster)
**Confidence:** 5

**Metareview:**

Reviewer Ratings - 2xBR and 2xBA

The paper proposes a CLIP driven method for facial expression recognition. The results are convincing and a detailed analysis has been performed. The rebuttal answers most of the queries.